# Estimating Nonlinear Neural Response Functions using GP Priors and Kronecker Methods

**Cristina Savin**
IST Austria
Klosterneuburg, AT 3400
csavin@ist.ac.at

**Gasper Tkačik**
IST Austria
Klosterneuburg, AT 3400
tkacik@ist.ac.at

## Abstract

Jointly characterizing neural responses in terms of several external variables promises novel insights into circuit function, but remains computationally prohibitive in practice. Here we use gaussian process (GP) priors and exploit recent advances in fast GP inference and learning based on Kronecker methods, to efficiently estimate multidimensional nonlinear tuning functions. Our estimator requires considerably less data than traditional methods and further provides principled uncertainty estimates. We apply these tools to hippocampal recordings during open field exploration and use them to characterize the joint dependence of CA1 responses on the position of the animal and several other variables, including the animal's speed, direction of motion, and network oscillations. Our results provide an unprecedentedly detailed quantification of the tuning of hippocampal neurons. The model's generality suggests that our approach can be used to estimate neural response properties in other brain regions.

## 1 Introduction

An important facet of neural data analysis concerns characterizing the tuning properties of neurons, defined as the average firing rate of a cell conditioned on the value of some external variables, for instance the orientation of an image patch in a V1 cell, or the position of the animal within an environment for hippocampal cells. As experiments become more complex and more naturalistic, the number of variables that modulate neural responses increases. These include not only experimentally targeted inputs but also variables that are no longer under the experimenter's control but which can be (to a certain extent) measured, either external (the behavior of the animal) or internal (attentional level, network oscillations, etc). Characterizing these complex dependencies is very difficult, yet it could provide important insights into neural circuits computation and function.

Traditional estimates of a cell's tuning properties often manipulate one variable at the time or consider simple dependencies between inputs and the neural responses e.g. Generalized Linear Models, GLM [1, 2]). There is comparatively little work that allows for complex input-output functional relationships on multidimensional input spaces [3–5]. The reasons for this are twofold. On one hand, dealing with complex nonlinearities is computationally challenging, on the other hand, constraints on experimental duration lead to a potentially very sparse sampling of the stimulus space, requiring additional assumptions for a sensible interpolation. This problem is further exacerbated in experiments in awake animals where the sampling of the stimulus space is driven by the animal behavior. The few solutions for nonlinear tuning properties rely on spline-based approximation of one-dimensional functions (for position on a linear track) [6] or assume a log-Gaussian Cox process generative model as a way to enforce smoothness of 2D functional maps [3–5]. These methods are usually restricted to at most two input dimensions (but see [4]).

Here we take advantage of recent advances in scaling GP inference and learning using Kronecker methods [7] to extend the approach in [3] to the multidimensional setting, while keeping the computational and memory requirements almost linear in dataset size $N$, $\mathcal{O}\left(dN^{\frac{d+1}{d}}\right)$ and $\mathcal{O}\left(dN^{\frac{2}{d}}\right)$, respectively (for $d$ dimensions) [8]. Our formulation requires a discretization of the input space,[1] but allows for a flexible selection of the kernels specifying different assumptions about the nature of the functional dependencies we are looking for in the data, with hyperparameters inferred by maximizing marginal likelihood. We deal with the non-gaussian likelihood in the traditional way by using a Laplace approximation of the posterior [8]. The critical ingredient for our approach is the particular form of the covariance matrix that decomposes into a Kronecker product over covariances corresponding to individual input dimensions, dramatically simplifying computations. The focus here is not on the methods per se but rather on their previously unacknowledged utility for estimating multidimensional nonlinear tuning functions.

The inferred tuning functions are probabilistic. The estimator is adaptive, in the sense that it relies strongly on the prior in regions of the input space where data is scarce, but can flexibly capture complex input-output relations where enough data is available. It naturally comes equipped with error bars which can be used for instance for detecting shifts in receptive field properties due to learning.

Using artificial data we show that inference and learning in our model can robustly recover the underlying structure of neural responses even in the experimentally realistic setting where the sampling of the input space is sparse and strongly non-uniform (due to stereotyped animal behavior). We further argue for the utility of spectral mixture kernels as a powerful tool for detecting complex functional relationships beyond simple smoothing/interpolation. We go beyond artificial data that follows the assumptions of the model exactly, and show robust estimation of tuning properties in several experimental recordings. For illustration purposes we focus here on data from the CA1 region of the hippocampus of rats, during an open field exploration task. We characterize several 3D tuning functions as a function of the animal's position but also additional internal (the overall activity in the network at the time) or external variables (speed or direction of motion, time within experiment) and use these to derive new insights into the distribution of spatial and non-spatial information at the level of CA1 principal cell activity.

## 2    Methods

**Generative model**

Given data in the form of spike count – input pairs $\mathcal{D} = \{y^{(i)}, \mathbf{x}^{(i)}\}_{i=1:N}$, we model neural activity as an inhomogeneous Poisson process with input-dependent firing rate $\lambda$ (as in [3], see. Fig. 1a):

$$\mathrm{P}(\mathbf{y}|\mathbf{x}) = \prod_i \mathrm{Poisson}\left(y^{(i)}; \lambda(\mathbf{x})^{(i)}\right), \qquad \text{where} \quad \mathrm{Poisson}\,(y; \lambda) = \frac{1}{y!}\lambda^y \mathrm{e}^{-\lambda}. \qquad (1)$$

The inputs $\mathbf{x}$ are defined on a $d$-dimensional lattice and the spike counts are measured within a time window $\delta t$ for which the input is roughly constant (25.6ms, given by the frequency of positional tracking).[2] We formalize assumptions about neural tuning as a GP prior $f \sim \mathcal{GP}(\mu, k_\beta)$, with $f = \log \lambda(\mathbf{x})$, with a constant mean $\mu_i = \alpha$ (for the overall scale of neural responses) and a covariance function $k(\cdot, \cdot)$ with hyperparameters $\beta$. This covariance function defines our assumptions about what kind of functional dependencies are expected in the data (smoothness, periodicity, etc.). The exponential linking $\mathbf{f}$ to $\lambda$ provides a mathematically convenient way to enforce positivity of the mean firing while keeping the posterior log-concave in $\mathbf{f}$, justifying the use of Laplace methods for approximating the posterior (see also [3]).

For computational tractability we restrict our model to the class of product kernels $k(\mathbf{x}, \mathbf{x}') = \prod_d k_d(x_d, x'_d)$ for which the covariance matrix decomposes as a Kronecker product $K = K_1 \otimes K_2 \otimes \ldots K_d$, allowing for efficient computation of determinants, matrix multiplications and eigen-decomposition in terms of the individual factors $K_i$ (see Suppl.Info. and [7]).

The individual kernels can be tailored to the specific application, allowing for a flexible characterization of individual input dimensions (inputs need not live in the same space, e.g. space-time, or can be

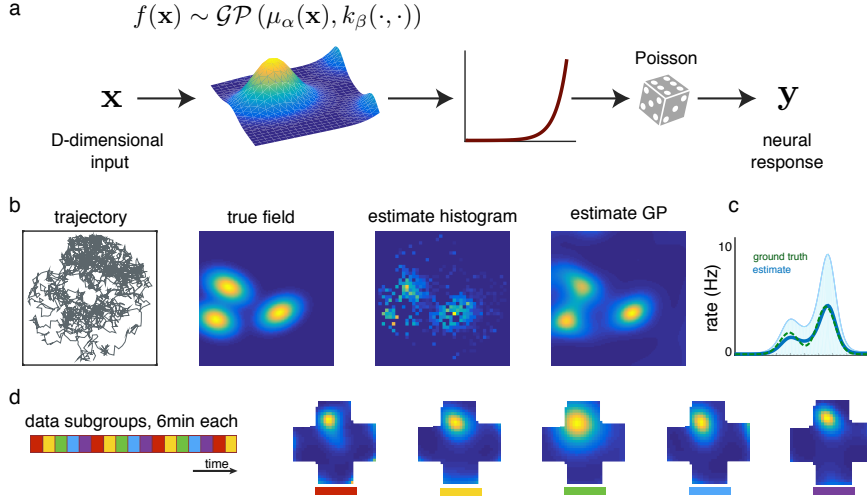

Figure 1: **Model overview and estimator validation.** a) Generative model: spike counts arise as Poisson draws with an input dependent mean, $f(\mathbf{x})$ with an exponential linkage function. b) A GP prior specifies the assumptions concerning the properties of this function (smoothness, periodicity, etc). c) Place field estimates from artificial data; left to right: the position of the animal modelled as a bounded random walk, ground-truth, traditional estimate (without smoothing), posterior mean of the inferred functional. d) Vertical slice through the posterior with shaded area showing the $2 \cdot \mathrm{sd}$ confidence region. d) Estimates of place field selectivity in an example CA1 recording during open field exploration in a cross-shaped box; separate estimates for 6min subsets.

periodic, e.g. the phase of theta oscillations). Here we use a classic squared-exponential (SE kernel for simple interpolation/smoothing tasks, $k_d(x, x') = \rho_d^2 \exp \frac{(x-x')^2}{2\sigma_d^2}$, with parameters $\beta = \{\boldsymbol{\rho}, \boldsymbol{\sigma}\}$ specifying the output variance and lengthscale [9]. For tasks involving extrapolation or discovering complex patterns we use spectral mixture (SM) kernels, as a powerful and mathematically tractable route towards automated kernel design [10]. SMs are stationary kernels defined as a linear mixture of basis functions in the spectral domain:

$$k_d(x, x') = \sum_{q=1}^{Q} w_q \exp\left(-2\pi^2 (x - x')^2 v_q\right) \cos(2\pi(x - x')\mu_q) \tag{2}$$

with parameters $\beta = \{\mathbf{w}, \boldsymbol{\mu}, \mathbf{v}\}$ defining the weights, spectral means and variances for each of the mixture components. Assuming $Q$ is large enough, such a spectral mixture can approximate any arbitrary kernel (the same way Gaussian mixtures can be used to approximate an arbitrary density). Moreover, many traditional kernels can be recovered as special cases; for instance the SE kernel corresponds to a single component spectral density with zero mean (see also [10]).

## Inference and learning

We sketch the main steps of the derivation here and provide the details in the Suppl. Info. Our goal is to find the hyperparameters $\theta = \{\alpha, \beta\}$ that maximize $\mathrm{P}(\theta|\mathrm{y}) \propto \mathrm{P}(\mathrm{y}|\theta) \cdot \mathrm{P}(\theta)$. We follow common practice in using a point estimate $\theta^* = \mathrm{argmax}_\theta \mathrm{P}(\theta|\mathrm{y})$ for the hyperparameters, and leave a fully probabilistic treatment to future work (e.g. using [11]). We use $\theta^*$ to infer a predictive distribution $\mathrm{P}(f^*|\mathcal{D}, \mathbf{x}^*, \theta^*)$ for a set of test inputs $\mathbf{x}^*$. Because of the Poisson observation noise these quantities do not have simple closed form solutions and some approximations are required. As it is customary [9], we use the Laplace method to approximate the log posterior $\log \mathrm{P}(\mathbf{f}|\mathcal{D}) = \log \mathrm{P}(\mathbf{y}|\mathbf{f}) + \log \mathrm{P}(\mathbf{f})$ with its second-order Taylor expansion around the maximum $\hat{\mathbf{f}}$. This results in a multivariate Gaussian approximate posterior, with mean $\hat{\mathbf{f}}$ and covariance $\left(H + K^{-1}\right)^{-1}$, where $H = -\nabla\nabla \log \mathrm{P}(\mathbf{y}|\mathbf{f})\,|_{\hat{\mathbf{f}}}$ is the Hessian of the log likelihood, and $K$ is the covariance matrix. Substituting the approximate posterior, we obtain the Laplace approximate marginal likelihood of the form:

$$\log(\mathbf{y}|\theta) = \log \mathrm{P}(\mathbf{y}|\hat{\mathbf{f}}) - 0.5\mathbf{z}'K^{-1}\mathbf{z} - 0.5\log|I + KH| \tag{3}$$

with $\mathbf{z} = K^{-1}(\hat{\mathbf{f}} - \boldsymbol{\mu})$. The approximate predictive distribution for $\theta^*$ is a multivariate Gaussian with mean $\mathbf{k}_* \nabla \log P(\mathbf{y}|\hat{\mathbf{f}})$ and covariance $k_{**} - \mathbf{k}_*' \left(H^{-1} + K\right)^{-1} \mathbf{k}_*$, where $k_*$ and $\mathbf{k}_{**}$ correspond to the test-data and test-test covariances, respectively [8]. Lastly the predicted tuning function for an individual test point $\lambda^*) = \exp(\mathbf{f}^*)$, is log-normal with closed-form expressions for mean and variance (see Suppl. Info.).

Standard methods for implementing these computations using the Cholesky decomposition require $\mathcal{O}\left(N^3\right)$ computations and $\mathcal{O}\left(N^2\right)$ memory, restricting their use to a few hundred data points. The efficient implementation proposed here relies on the Kronecker structure of the covariance matrix (which makes eigenvalue decomposition and matrix vector products very fast, see Suppl.Info.), with linear conjugate gradients optimization and a lower bound on the marginal likelihood for hyperparameter learning. The predictive distribution can be efficiently evaluated in $\mathcal{O}\left(dN^{\frac{d+1}{d}}\right)$ (with a hidden constant given by the number of Newton steps needed for convergence, cf. [8]) Our implementation is based on the gpml library [9] and the code is available online. A more detailed description of the algorithmic details is provided in the Suppl. Info. In practice, this means that it takes minutes on a laptop to estimate a 3D field for a 30min dataset (2-5min depending on the coarseness of the grid), with a traditional 2D field estimated in 20-30sec.

## 3   Results

**Estimator validation**

We first validated our implementation on artificial data with known statistics.[3] We defined a circular arena with 1m diameter and simulated the animal's behavior as a random walk with reflective bounds (Fig. 1 b, left panel). This random process would eventually uniformly cover the space, but for short sessions it yields occupancy maps similar to those seen in real data. We calibrated diffusion parameters to roughly match CA1 statistics (average speed 5cm/sec, peak firing 5-10Hz, 10-30min long sessions). Inferring the underlying place field was already robust with 10min sessions, with the posterior mean $\mathbf{f}^*$ close to the ground truth (SE kernel, see Fig. 1 c). In comparison, the traditional histogram-based estimates is quite poor (Fig. 1 b, left panel), though it can potentially be improved by gaussian smoothing at the right spatial scale (although not without caveats, see Suppl. Info.).

It is more difficult to quantify the effects of the various approximations on real data where the assumptions of the model are not matched exactly. Our approach was to check the robustness of the GP-based estimates on subsets of the data constructed by combining every 5th data point (see left panel in Fig. 1 d). This partitioning was designed to ensure that subsets are as statistically similar as possible, sharing slow fluctuations in responses (e.g. due to variations in attentional levels, or changes in behavior). An example cell's response is shown in Fig. 1 d. Our analysis revealed robust field estimation in most cells, provided they were reasonably active during the session (with mean firing rates >0.1Hz; we discarded the non-responsive cells from subsequent analyses).

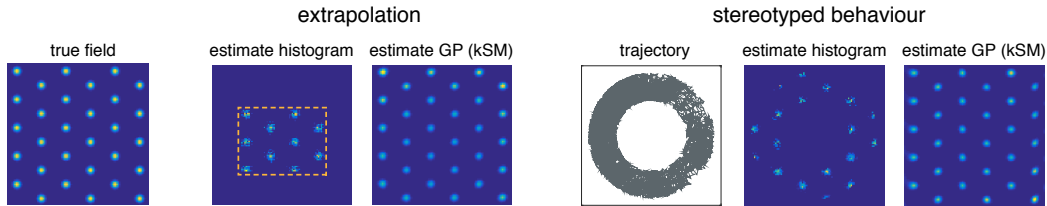

Figure 2: **Spectral mixture kernels for modelling complex structure.** We use artificial data with hexagonal grid structure mimicking MEC responses. Extrapolation task: the animal's position is restricted to the orange delimited region of the environment. Stereotyped behavior: the simulated animal performs a bounded random walk within an annulus . In both cases, we recover the full field, beyond these borders (GP estimate) using a spectral mixture kernel (kSM).

**Spectral mixture kernels for complex functional dependencies**

Place field estimation is relatively easy in a traditional open field exploration session (30min). The main challenge is getting robust estimates on the time scale of a few minutes (e.g. in order to be able to detect changes due to learning), which we have seen a GP-based estimator can do well. A much more difficult problem is detecting tuning properties in a cheeseboard memory task [12]. What distinguishes this setup is that fact that the animal quickly discovers the location of the wells containing rewards, after which its running patterns become highly stereotypical, close to the shortest path that traverses the reward locations. While it is hard to figure out place field selectivity for locations that the animal never visits, GP-based estimators may have an advantage compared to traditional methods when functional dependencies are structured, as is the case for grid cells in the medial enthorinal cortex (MEC) [13, 14]. When tuning properties are complex and structured we can exploit the expressive power of spectral mixture kernels (SM) to make the most of very limited data.

We simulated two versions of this scenario. First, we defined an extrapolation task in which the animal's behaviour is restricted to a subregion of the environment (marked by orange lines in the 2nd panel of Fig. 2) but we want to infer the spatial selectivity outside these borders. The second scenario attempts to mimic the animal running patterns in a cheeseboard maze (after learning) by restricting the trajectory within a ring (random walk with reflective boundaries in both cases). Using a 5 component spectral mixture kernel we were able to fully reconstruct the hexagonal lattice structure of the true field despite the size of the observed region covering only about 2 times the length scale of the periodic pattern. In contrast, traditional methods (including GP-based inference with standard SE kernels) would fail completely at such extrapolation. While such complex patterns of spatial dependence are restricted to MEC (and the estimator is probably best suited for ventral MEC, where grids have a small length scale [15]) it is conceivable that such extrapolation may also be useful in the temporal domain, or more generally for cortical responses in neurons which have so far eluded a simple functional characterization.

**Spatial and non-spatial modulation of CA1 responses**

To explore the multidimensional characterization of principal cell responses in CA1 we constructed several 3D estimators where the input combines the position of the animal within a 2D environment with an additional non-spatial variable.[4] The first non-spatial variable we considered is the network state, quantified as the population spike count, $k = \sum_{i=1}^{N_{\mathrm{neurons}}} y_i$ (naturally a discrete variable between 0 and some $k_{\max}$). This quantity provides a computationally convenient proxy for network oscillations and has been recently used in a series of studies on the statistics of population activity in the retina and cortex [16–19]. Second, we considered the animal's speed and direction of motion (with a coarse discretization), motivated by past work on non-spatial modulation of place fields on linear tracks [20]. Third, we also considered input variable $t$ measuring time within a session (SE kernel; 3-5 min windows), as a way to examine the stability of spatial tuning over time. For all analyses, positional information was discretized on a $32 \times 32$ grid, corresponding to a spacing of 2.5cm, comparable to the binning resolution used in traditional place field estimates. The animal speed (estimated from the positional information with 250ms temporal smoothing) varied between 0 and about 25cm/sec, with a very skewed distribution (not shown). Small to medium variations in the coarseness of the discretization did not qualitatively affect the results although the choice of prior becomes more important on the tail of the speed distribution, where data is scarce.

The resulting 3D tuning functions are shown in Fig. 3 for a few example neurons. First, network state modulates the place field selectivity in most CA1 neurons in our recordings. The typical modulation pattern is a monotonic increase in firing with $k$ (Fig. 3, a, top), although we also found $k$-dependent flickering in a minority of the cells (Fig. 3a, middle), and very rarely $k$ invariance (Fig. 3a, bottom). Rate remapping is also the dominant pattern of speed-dependent modulation in our data set (Fig. 3b). In terms of place field stability over time, about half the cells were stable during a 30min session in a familiar environment, with occasionally higher firing rates at the very beginning of the trial (Fig. 3c, top), while the rest showed fluctuation in representations (Fig. 3c, bottom). Results shown for 5min windows, but results very similar for 3min.

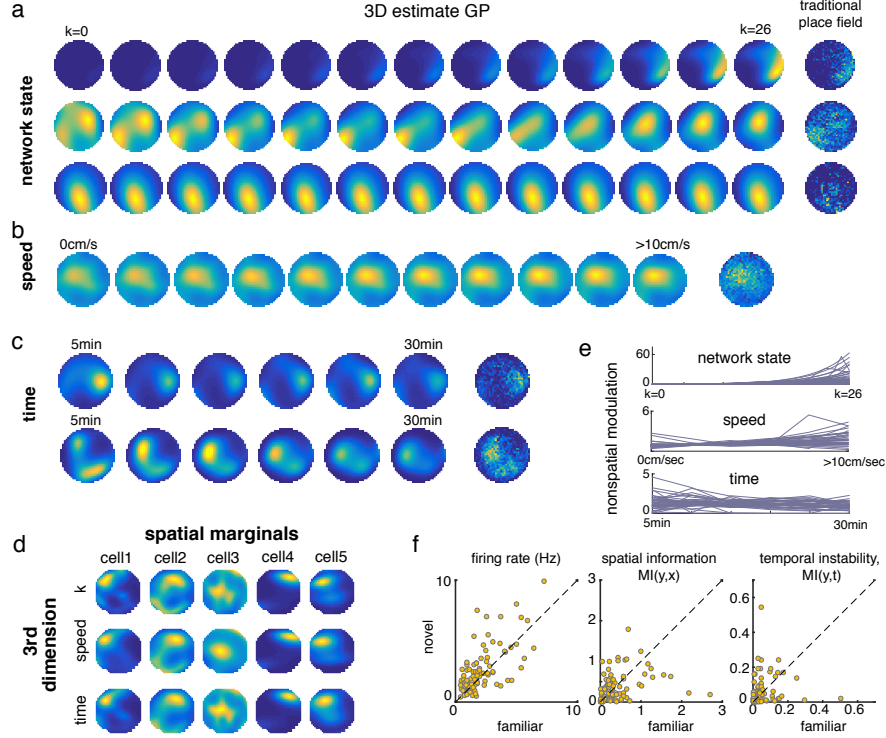

Figure 3: **Estimating 3D response dependences in CA1 cells.** a) Conditional place fields when constraining the network state, defined by the average population activity $k$. c) Conditional place fields as a function of the time within a 30min session, used to assess the stability of the representation. In all cases, the rightmost field corresponds to the traditional place field ignoring the 3rd dimension. d) Sanity check: marginal statistics of the place field selectivity obtained independently from the 3D fields in 5 example cells. e) Population summary of the degree of modulation of spatial selectivity by non-spatial variables; see text for details. f) Within comparison of cell properties during the exploration of a familiar vs. a novel environment.

As a sanity check of our 3D estimators' quality, we independently computed the traditional place field by marginalizing out the 3rd dimension for each of our 3D estimates. We used the empirical distribution as a prior for the non-spatial dimensions, and an uniform prior for space. Reassuringly we find that the estimates computed after marginalization are very close to the simple 2D place field map in all but 2 cells, which we exclude from the next analysis (examples in Fig. 3d). This provides additional confidence in the robustness of the estimator in the multidimensional case.

Since we have a closed form expression for the map between stimulus dimensions and neural responses, we can estimate the mutual information between neural activity and various input variables as a way to dissect their contribution to coding. First, we visualize the modulation of spatial selectivity by the non-spatial variable as the spatial information conditioned on the 3rd variable, normalized by the marginal spatial information, $\frac{\mathrm{MI(x,y|z)}}{\mathrm{MI(x,y)}}$, with $z$ generically denoting any of the non-spatial variables (approximate closed form expression given $\mathbf{f}$ and Poisson observation noise). We see monotonic increases in spatial information with $k$ (Fig. 3e, top), and speed (Fig. 3e, top) at the level of the population, and a weak decrease in spatial information over time (possibly due to higher speeds at the beginning of the session, combined with heightened attention/motivation levels). In terms of the division of spatial vs. non-spatial information across cells, we found that space selective cells have weaker $k$-modulation (Spearman $\mathrm{corr}(\mathrm{MI}(y,x),\mathrm{MI}(y,k) = -0.17)$. This however does not exclude the possibility that theta-coupled cells have additional spatial information at the fine temporal scale. Additionally, there is little correlation between the coding of position and speed ($\mathrm{corr}(\mathrm{MI}(y,x),\mathrm{MI}(y,\mathrm{speed})) = -0.03$), suggesting that the encoding of the two is relatively orthogonal at the level of the population. Somewhat unexpectedly, we found a cell's temporal stability to be largely independent of its spatial selectivity $\mathrm{corr}(\mathrm{MI}(y,x),\mathrm{MI}(y,t)) = -0.04$.

Motivated by recent observations that the overall excitability of cells may be predictive of both their spatial selectivity and of the rigidity of their representation [21], we compared the overall firing rate of the cells with their spatial and non-spatial selectivity. We found relatively strong dependencies, with positive correlations between firing rate and spatial information ($cc = 0.21$), network influence ($cc = 0.43$) and the cell's stability ($cc = 0.38$). When comparing these quantities in the same cells as the animal visits a familiar or a novel environment (93 cells, 20min in each environment) we found additional nontrivial dependences between spatial and non-spatial tuning. Although the overall firing rates of the cells are remarkably preserved across conditions (reflecting general cell excitability, $cc = 0.66$), the subpopulation of cells with strong spatial selectivity is largely non-overlapping across environments ($\mathrm{corr}(\mathrm{MI}_{\mathrm{fam}}(y,x), \mathrm{MI}_{nov}(y,x) = 0.07$). Moreover, the temporal stability of the representation is also environment specific ($\mathrm{corr}(\mathrm{MI}_{\mathrm{fam}}(y,t), \mathrm{MI}_{nov}(y,t) = -0.04$). Overall, these results paint a complex picture of hippocampal coding, the implications of which need further empirical and theoretical investigation.

Lastly, we studied the dependence of CA1 responses on the animal's direction of motion. Although directional selectivity is well documented on a linear track [20] it remains unclear if a similar behavior occurs in a 2D environment. The main challenge comes from the poor sampling of the position×direction-of-motion input space, something which our methods can handle readily. To construct directionally selective place field estimates in 2D we took inspiration from recent analyses of 2D phase procession [22] conditioning the responses on the main direction of motion within the place field. Specifically, we used our estimation of a traditional 2D place field to define a region of interest (ROI) that covers 90% of the field for each cell (Fig. 4. We isolated all trajectory segments that traverse this ROI and classified them based on the primary direction of motion along the cardinal orientations. We then computed place field estimates for each direction, with data outside the ROI shared across conditions. To avoid artefacts due to the stereotypical pattern of running along the box borders, we restricted this analysis to cells with fields in the central part of the environment (10 cells). A set of representative examples for the resulting directional fields are shown in Fig. 4d. We found the fields to be largely invariant to direction of motion in our setup, with small displacements in peak firing possibly due to differences between the perceived vs. the camera-based measurements of position (see also [22]). Overall, these results suggest that, in contrast to linear track behavior, CA1 responses are largely invariant to the direction of motion in an open field exploration task.

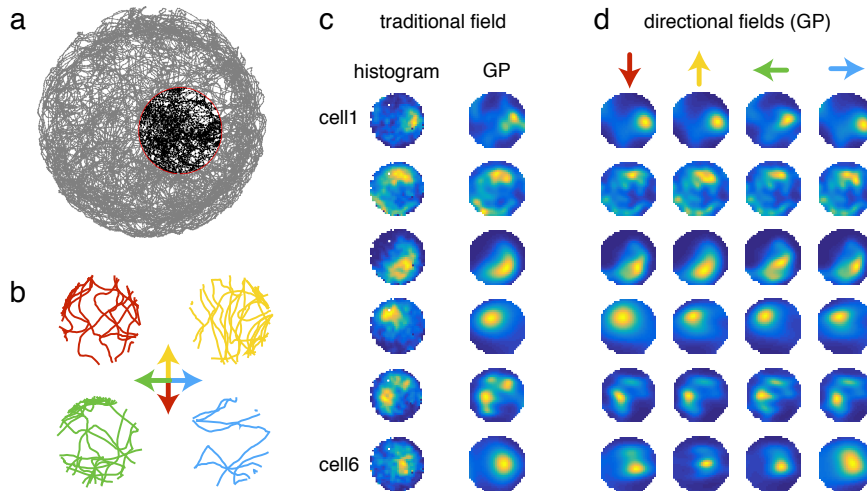

Figure 4: **Directional selectivity in CA1 cells.** a) Cell specific ROI that covers the classic place field (example corresponding to cell 6). b) Classification of the traversals of the region of interest as a function of the primary direction of motion along the cardinal directions. Out of ROI data shared across conditions. c) Traditional place field estimates for example CA1 cells and d) their corresponding direction-specific tuning.

## 4 Discussion

Strong constraints on experiment duration, poor sampling of the stimulus space and additional sources of variability that are not under direct experimental control make the estimation of tuning properties during awake behavior particularly challenging. Here we have shown that recent advances on fast GP inference based on Kronecker methods allow for a robust characterization of multidimensional non-linear tuning functions, which was inaccessible to traditional methods. Furthermore, our estimators inherit all the advantages of a probabilistic approach, including a principled way of dealing with the non-uniform sampling of the input space and natural uncertainty estimates.

Our methods can robustly estimate place fields with one order of magnitude fewer data points. Furthermore, they allow for more than two-dimensional inputs. While one could imagine it would suffice to estimate separate place fields conditioned on each value of the non-spatial dimension, $z$, the joint estimator has the advantage that it allows for smoothing across $z$ values, borrowing strength from well-sampled regions of the $z$ space to make better estimates for poorly sampled $z$ values.

Several related algorithms have been proposed in the literature [3–5], which vary primarily in how they handle the tradeoff between kernel flexibility and the computational time required for inference and learning (see Table 1). At one extreme, [3] strongly restricts the nature of the covariance matrix to nearest-neighbour interactions on a 2D grid (resulting in a band-diagonal inverse covariance matrix) which allows them to exploit sparse matrix techniques to estimate the posterior mean in linear time. At the other extreme, [4, 5] allow for an arbitrary covariance structure, but are computationally prohibitive, $\mathcal{O}\left(N^3\right)$. Our proposal sits between these extremes in that it achieves close-to-linear computational and memory costs without significantly restricting the flexibility of the covariance structure (for a better intuition of the effect of different covariances, see also Fig. S1). In particular, it can be combined with powerful spectral mixture kernels to extract complex functional dependencies that go beyond simple smoothing. This opens the door to a variety of previously inaccessible tasks such as extrapolation. Moreover, it allows for an agnostic exploration of the neural responses functional space, which could be used to discover novel tuning properties in cells for which coding is poorly understood.

When applied to CA1 data, our multidimensional estimators revealed a complex picture of the modulation of neural responses by spatial and non-spatial inputs in the hippocampus. First we confirmed linear track results concerning the speed and oscillatory modulation of spatial tuning. Furthermore, we revealed additional insights into the interaction between the representation of space and these non-spatial dimensions, which go beyond the capabilities of traditional methods. Most notably we found 1) a mostly orthogonal representation of speed and position, that 2) place field stability cannot be easily explained in terms of cell excitability or spatial selectivity, although 3) it is environment specific. Lastly, while we showed 2D place field maps to be direction-invariant in an open field exploration task, more interesting directional dependencies may be revealed in other 2D tasks, where the direction of motion is behavioraly more relevant (e.g. cheeseboard). Importantly, there is nothing hippocampus-specific in the methodology. Hence fast GP inference using Kronecker methods, combined with expressive kernels, may provide a general-purpose tool for characterizing neural responses across brain regions.

Table 1: Summary comparison of different estimators.

| Algorithm | Kernel function | Computing cost | Memory cost | Data size |
|---|---|---|---|---|
| Rad et al. 2010 [3] | sparse banded inverse co-variance | $\mathcal{O}\left(N\right)$ | $\mathcal{O}\left(N\right)$ | $10^5$ |
| Park et al. 2014 [4] | SE, any in principle | $\mathcal{O}\left(N^3\right)$ | $\mathcal{O}\left(N^2\right)$ | $< 10^3$ |
| Savin & Tkacik | SE and SM, works for any tensor-product | $\mathcal{O}\left(dN^{\frac{d+1}{d}}\right)$ | $\mathcal{O}\left(dN^{\frac{2}{d}}\right)$ | $10^5$ |

**Acknowledgments**

We thank Jozsef Csicsvari for kindly sharing the CA1 data. This work was supported by the People Programme (Marie Curie Actions) of the European Union's Seventh Framework Programme (FP7/2007-2013) under REA grant agreement no. 291734.

## Footnotes

[1]In practice many input dimensions are discrete to begin with (e.g. measurements of an animal's position), so this is a weak requirement. The coarseness of the discretization depends on the application.

[2]Input noise is ignored here, but could be explicitly incorporated in the generative model [9].

[3]Here we show a 2D example for simplicity; we obtained very similar results with 3D artificial inputs.

[4]We chose to estimate multiple 3D fields rather than jointly conditioning on all variables mainly for simplicity; this strategy has the added bonus of providing sanity checks for the quality of the different estimates.

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
