[Supplementary Material]

# Estimating Nonlinear Neural Response Functions using GP Priors and Kronecker Methods – Supplementary Information –

**Cristina Savin**
IST Austria
Klosterneuburg, AT 3400
csavin@ist.ac.at

**Gasper Tkačik**
IST Austria
Klosterneuburg, AT 3400
tkacik@ist.ac.at

The derivations below rely strongly on [1–4]. We provide a summary here for reading convenience and refer the reader to the original references for further details. The code for our implementation is available online (https://crissavin@bitbucket.org/crissavin/pfgp.git).

## S1  Inference and learning

The generative model of the data takes the form of a log-normal Cox process, where spike counts $\mathbf{y}_i$ are generated by an inhomogeneous Poisson process with input-dependent mean $\lambda_i = \lambda\left(\mathbf{x}^{(i)}\right)$:

$$P(\mathbf{y}|\mathbf{x}) = \prod_i \text{Poisson}\left(y_i; \lambda_i\right), \qquad \text{where} \quad \text{Poisson}\left(y; \lambda\right) = \frac{1}{y!}\lambda^y \mathrm{e}^{-\lambda}. \tag{1}$$

where the inputs $\mathbf{x}$ are defined on a $d$-dimensional lattice and the spike counts are measured within a time window $\delta t$. Assumptions concerning the nature of the functional dependency linking inputs to outputs is formalized as a gaussian process (GP) prior for the log-firing rate $f = \log \lambda(\mathbf{x}) \sim \mathcal{GP}(\mu, k_\beta)$ with constant mean $\mu_i = \alpha$ and covariance function $k(\cdot, \cdot)$ specified by hyperparameters $\beta$. The choice of covariance kernel determines the flexibility of the model and the computational complexity of the implementation. We postpone the discussion on the tradeoffs associated with different covariances for the moment, and focus on inference and learning in the general case.

First, the posterior over latent variable $\mathbf{f}$ given the data $\mathcal{D} = \{y_i, \mathbf{x}_i\}_{i=1:N}$ for fixed hyperparameters $\theta$[1] takes the form:

$$P\left(\mathbf{f}|\mathcal{D}\right) = \frac{P\left(\mathbf{y}|\mathbf{f}\right) P\left(\mathbf{f}|\mathbf{x}\right)}{P\left(\mathbf{y}|\mathbf{x}\right)} \tag{2}$$

where $P\left(\mathbf{y}|\mathbf{f}\right) = \prod_i P\left(y_i|f_i\right)$ and $P\left(\mathbf{f}|\mathbf{x}\right) = \mathcal{N}(\mathbf{f}|\boldsymbol{\mu}, \mathbf{K})$, where $\boldsymbol{\mu}$ and $\mathbf{K}$ are the GP mean and covariance for inputs $\mathbf{x}$. Because of the Poisson likelihood this expression does not have a simple closed form, but can be approximated with a normal distribution using the Laplace method [1].

In more detail, the Laplace approximation entails a 2nd order Taylor expansion of $\log P\left(\mathbf{f}|\mathcal{D}\right)$ around the maximum of the posterior $\hat{\mathrm{f}} = \text{argmax}_{\mathbf{f}} P\left(\mathbf{f}|\mathbf{y}, \mathbf{x}\right)$, resulting in an approximate posterior:

$$q\left(\mathbf{f}|\mathbf{x}, \mathbf{y}\right) = \mathcal{N}\left(\mathbf{f}; \hat{\mathbf{f}}, \left(\mathbf{K} + \mathbf{H}^{-1}\right)^{-1}\right) \tag{3}$$

where the Hessian of the log-likelihood $\mathbf{H} = -\nabla\nabla \log P(\mathbf{y}|\mathbf{f})\left.\right|_{\hat{\mathbf{f}}}$ is diagonal with $H_{ii} = \exp(\hat{f}_i)$ since the noise is conditionally independent Poisson given $\mathbf{f}$. Our linkage function is convex and log-concave so the posterior has a unique maximum $\hat{\mathbf{f}}$ which can be computed numerically as the

maximum of unnormalized log-posterior $\Psi(\mathbf{f}) = \log P(\mathbf{y}|\mathbf{f}) + \log P(\mathbf{f}|\mathbf{x})$, via Newton's method. This involves updates of the form (implementation details follow below):

$$\mathbf{f}^{\text{new}} \leftarrow \mathbf{f}^{\text{old}} + (\nabla\nabla\Psi)^{-1}\nabla\Psi. \tag{4}$$

The hyperparameters $\theta = \{\alpha, \beta\}$ are learned by maximizing the marginal likelihood: [2]

$$P(\mathbf{y}|\mathbf{x}) = \int P(\mathbf{y}|\mathbf{f})P(\mathbf{f}|\mathbf{x})d\mathbf{f} = \int \exp(\Psi(\mathbf{f}))\,d\mathbf{f}. \tag{5}$$

This integral can be computed analytically using the same Laplace approximation,

$$\Psi(\mathbf{f}) \approx \Psi\left(\hat{\mathbf{f}}\right) - \frac{1}{2}\left(\mathbf{f} - \hat{\mathbf{f}}\right)^{\text{T}} \cdot \left(\mathbf{K} + \mathbf{H}^{-1}\right)^{-1} \cdot \left(\mathbf{f} - \hat{\mathbf{f}}\right) \tag{6}$$

resulting in the log marginal likelihood:

$$\mathcal{L}(\Theta) = -\frac{1}{2}\hat{\mathbf{f}}^{\text{T}}\mathbf{K}^{-1}\hat{\mathbf{f}} + \log P\left(\mathbf{y}|\hat{\mathbf{f}}\right) + \log|\mathbf{I} + \mathbf{K}\mathbf{H}|. \tag{7}$$

Consistent with past approaches, we use $\theta^* = \operatorname{argmax}_\theta P(\theta|\mathbf{y})$ to infer the tuning function for a set of test inputs $\mathbf{x}^*$.

Predicting the tuning function at a set of test locations $\mathbf{x}^*$ requires computing the mean and covariance of latent $\mathbf{f}^*$, after marginalizing $\mathbf{f}$ under the normal approximation $q$:

$$\mathbb{E}_q[\mathbf{f}^*|\mathcal{D}, \mathbf{x}^*] = \alpha + K(\mathbf{x}^*, \mathbf{x}) \cdot \mathbf{K}^{-1} \cdot \hat{\mathbf{f}} = \alpha + K(\mathbf{x}^*, \mathbf{x}) \cdot \nabla P(\mathbf{y}|\mathbf{f})|_{\hat{\mathbf{f}}} \tag{8}$$

$$\mathbb{V}_q[\mathbf{f}^*|\mathcal{D}, \mathbf{x}^*] = K(\mathbf{x}^*, \mathbf{x}^*) - K(\mathbf{x}^*, \mathbf{x}) \cdot (\mathbf{K} + \mathbf{H}^{-1})^{-1} \cdot K(\mathbf{x}, \mathbf{x}^*). \tag{9}$$

Since we have used an exponential linkage function, the predicted tuning function for an individual test point $\lambda(\mathbf{x}^*) = \exp(\mathbf{f}^*)$, is log-normal with mean $\exp(\mu_{f^*} + \sigma_{f^*}^2/2)$ and variance $\left(\exp(\sigma_{f^*}^2) - 1\right)\exp\left(2\mu_{f^*} + \sigma_{f^*}^2\right)$, where $\mu_{f^*} = \mathbb{E}_q[f^*|\mathcal{D}, \mathbf{x}^*]$ and $\sigma_{f^*}^2 = \mathbb{V}_q[f^*|\mathcal{D}, \mathbf{x}^*]$.

## S2 Efficient implementation using Kronecker methods

As we have seen above, numerical optimization is required for determining $\mathbf{f}^*$ and $\theta^*$. This involves solving linear systems and determinants for $N \times N$ matrices, often via Cholesky decomposition, with computational cost $\mathcal{O}(N^3)$ and memory cost $\mathcal{O}(N^2)$, respectively, in the general case. These computations can be substantially sped up if additional assumptions are made concerning the inputs and the structure of $\mathbf{K}$. Here we rely on the fact that the inputs $\mathbf{x_i}$ lie on a (non-uniform) cartesian grid with arbitrary elements $x_i \in \mathbb{R}^{D_i}$ where $D_i$ is the size of the $i$-th grid dimension (for simplicity we take these to all be the same in the following). Second, we assume that the prior covariance is described by a *tensor product kernel*, meaning that it can be written as a product of (arbitrary) kernels operating on scalar input spaces:

$$k(\mathbf{x}, \mathbf{y}) = \prod_{i=1}^d k_i(x_i, y_i). \tag{10}$$

Tensor products kernel seem a natural choice when different input dimensions live in different spaces with specific functional dependencies (e.g. for space-time data we may want to assume smoothness over space and periodicity in time). While not all popular kernels fall into this category, many of them do. For instance the popular squared-exponential kernel can be written in this form. The particular form of a tensor product kernel implies that the covariance matrix has Kronecker structure, i.e. it can be written as a Kronecker product

$$\mathbf{K} = \bigotimes_i \mathbf{K}_i \tag{11}$$

This dramatically simplifies computations.

## S2.1 Kronecker product properties

A detailed explanation of Kronecker algebra properties is beyond the scope of this text, but we'll review here the most relevant features (see also chapter 5 and corresponding appendix from [2]). Briefly, the Kronecker product of two matrices $\mathbf{A}$ and $\mathbf{B}$ of size $m \times n$ and $p \times q$, respectively, is a $mp \times nq$ matrix of the form:

$$\mathbf{A} \otimes \mathbf{B} = \begin{bmatrix} a_{11}\mathbf{B} & \cdots & a_{11}\mathbf{B} \\ \vdots & \ddots & \vdots \\ a_{m1}\mathbf{B} & \cdots & a_{mn}\mathbf{B} \end{bmatrix} \tag{12}$$

Because of this particular structure, traditional matrix operations on $\mathbf{K}$ can be rewritten in terms of operations on the (much smaller) component matrices $\mathbf{K}_i$:

$$\text{Transpose}: \qquad \mathbf{K}^{\mathrm{T}} = \bigotimes_i \mathbf{K}_i^{\mathrm{T}} \tag{13}$$

$$\text{Inverse}: \qquad \mathbf{K}^{-1} = \bigotimes_i \mathbf{K}_i^{-1} \tag{14}$$

$$\text{Eigen} - \text{decomposition}: \quad \mathbf{K} = \bigotimes_i \mathbf{Q}_i \cdot \bigotimes_i \mathbf{\Lambda}_i \cdot \bigotimes_i \mathbf{Q}_i^{\mathrm{T}} \tag{15}$$

$$\text{Determinants}: \qquad \det(\mathbf{K}) = \prod_i \det(\mathbf{K}_i)^{D_i} \tag{16}$$

$$\text{Trace}: \qquad \operatorname{tr}(\mathbf{K}) = \prod_i \operatorname{tr}(\mathbf{K}_i) \tag{17}$$

$$\text{Vec}: \quad \operatorname{vec}(\mathbf{AXB}) = \mathbf{A} \otimes \mathbf{B} \operatorname{vec}(\mathbf{X}) \tag{18}$$

where the eigen-decomposition of the individual components is $\mathbf{K}_i = \mathbf{Q}_i \mathbf{\Lambda}_i \mathbf{Q}_i^{\mathrm{T}}$ and $\operatorname{vec}(\mathbf{X})$ is a vector obtained by concatenating column-wise the elements in matrix $\mathbf{X}$.

Fast matrix-vector products (needed for the Newton updates) can be computed using tensor algebra (based on Eq.18, see Suppl. Info. in [3] for full derivation):

$$\left(\bigotimes_i \mathbf{K}_i\right) \cdot \mathbf{u} = \operatorname{vec}\left([\mathbf{K}_1, \dots [\mathbf{K}_{d-1}, [\mathbf{K}_d, \mathbf{U}]]]\right) \stackrel{\text{def}}{=} \operatorname{kronmvprod}(\mathbf{K}_1, \dots \mathbf{K}_d, \mathbf{u}) \tag{19}$$

where the brackets denote a matrix product, followed by the reshaping of the resulting matrix, $[\mathbf{K}_d, \mathbf{U}] = \operatorname{reshape}\left(\mathbf{K}_d\mathbf{U}, N^{\frac{1}{d}}, N^{\frac{d-1}{d}}\right)$ and $\mathbf{U} = \operatorname{reshape}\left(\mathbf{u}, N^{\frac{1}{d}}, N^{\frac{d-1}{d}}\right)$.

## S2.2 Kronecker GP implementation

For numerical stability, the update equations for Newton's method are first rewritten in the form [1]:

$$\mathbf{f}^{\mathrm{new}} \leftarrow \mathbf{Ka} \tag{20}$$

using the sequence of transformations:

$$\mathbf{B} = \mathbf{I} + \mathbf{H}^{\frac{1}{2}}\mathbf{K}\mathbf{H}^{\frac{1}{2}} \tag{21}$$

$$\mathbf{Q} = \mathbf{H}^{\frac{1}{2}}\mathbf{B}^{-1}\mathbf{H}^{\frac{1}{2}} \tag{22}$$

$$\mathbf{b} = \mathbf{H}(\mathbf{f} - \boldsymbol{\mu}) + \nabla\mathrm{P}(\mathbf{y}|\mathbf{f}) \tag{23}$$

$$\mathbf{a} = \mathbf{b} - \mathbf{QKb} \tag{24}$$

After the variable change we still need to invert $\mathbf{B}$, which unfortunately does not have Kronecker structure. This problem can be circumvented by using linear conjugate gradients (LCG), an iterative method that replaces the costly inversion with matrix-vector products. In this representation, the variance of the predictive distribution $\mathrm{P}(\mathbf{f}^*|\mathcal{D}, \mathbf{x}^*, \theta)$ becomes:

$$\mathbb{V}_q[\mathbf{f}^*|\mathcal{D}, \mathbf{x}^*] = \mathbf{K}_{**} - \mathbf{K}_* \mathbf{Q} \mathbf{K}_*^{\mathrm{T}}. \tag{25}$$

Evaluating the marginal likelihood requires computing $\log|\mathbf{I} + \mathbf{KH}|$. Using the eigenvalue decomposition of $\mathbf{K}$, obtained in $\mathcal{O}\left(dN^{\frac{3}{d}}\right)$, one can bound this complexity term (using Fiedler's bound for the log determinant of a sum of Hermitian positive semidefinite matrices, see [4] for details) by:

$$\log|\mathbf{I} + \mathbf{KH}| \leq \sum_i \log(1 + e_i h_i) \tag{26}$$

where $e_i$ and $h_i$ are the eigenvalues of $\mathbf{K}$ and $\mathbf{H}$, respectively.[3] This results in a lower bound for the marginal likelihood of the form:

$$\mathcal{L}(\theta) \geq -\frac{1}{2}\hat{\mathbf{f}}^{\mathrm{T}}\mathbf{K}^{-1}\hat{\mathbf{f}} + \log \mathrm{P}\left(\mathbf{y}|\hat{\mathbf{f}}\right) - \frac{1}{2}\sum_i \log(1 + e_i h_i) \tag{27}$$

where the inversion of $\mathbf{K}$ can be done efficiently using Eq.14.

Pseudocode for the final algorithm is reproduced from [4] as Algorithm 1. For simplicity, we assume here that the grid has the same cardinality across all dimensions, i.e. $D_i = N^{\frac{1}{d}}$, for all $i$. Given this assumption, matrix-vector products (function kronmvprod) can be evaluated in $\mathcal{O}\left(dN^{\frac{d+1}{d}}\right)$. Consequently, computing $\mathbf{f}^*$ requires $\mathcal{O}\left(mdN^{\frac{d+1}{d}}\right)$ operations, where $m$ is the number of Newton steps needed for convergence. Memory-wise, we only need to store the matrices $\mathbf{K}_i$ with total size $\mathcal{O}\left(dN^{\frac{2}{d}}\right)$.

---

**Algorithm 1** Kronecker GP inference and learning

1: **procedure** $\mathrm{GP}(\theta, \boldsymbol{\mu}, \mathbf{K}_{1:d}, \mathrm{P}(\mathbf{y}|\mathbf{f}), \mathbf{y})$
2: $\quad \boldsymbol{\alpha} \leftarrow 0$
3: $\quad$ **repeat**
4: $\quad\quad \mathbf{f} \leftarrow \mathbf{K}\boldsymbol{\alpha}$ $\hfill \triangleright$ using kronmvprod
5: $\quad\quad \mathbf{H} \leftarrow -\nabla\nabla\mathrm{P}(\mathbf{y}|\mathbf{f})$ $\hfill \triangleright$ diagonal matrix
6: $\quad\quad \mathbf{b} = \mathbf{H}\left(\mathbf{f} - \boldsymbol{\mu}\right) + \nabla\mathrm{P}(\mathbf{y}|\mathbf{f})$
7: $\quad\quad \mathbf{z} \leftarrow \mathrm{CG}(\mathbf{B}, \mathbf{H})$ $\hfill \triangleright$ use conjugate gradient to solve $\mathbf{Bz} = \mathbf{H}^{-\frac{1}{2}}\mathbf{b}$
8: $\quad\quad \Delta\boldsymbol{\alpha} \leftarrow \mathbf{H}^{\frac{1}{2}}\mathbf{z} - \boldsymbol{\alpha}$
9: $\quad\quad \hat{\xi} \leftarrow \mathrm{argmin}_\xi \Psi(\boldsymbol{\alpha} + \xi\Delta\boldsymbol{\alpha})$ $\hfill \triangleright$ line search
10: $\quad\quad \boldsymbol{\alpha} \leftarrow \boldsymbol{\alpha} + \hat{\xi}\Delta\boldsymbol{\alpha}$ $\hfill \triangleright$ actual update step
11: $\quad$ **until** convergence
12: $\quad \mathbf{e} \leftarrow \mathrm{eig}(\mathbf{K})$ $\hfill \triangleright$ exploit Kronecker structure
13: $\quad Z =\leftarrow \frac{1}{2}\boldsymbol{\alpha}^{\mathrm{T}}(\mathbf{f} - \boldsymbol{\mu}) + \frac{1}{2}\sum_i \log(1 + e_i h_i) - \log\mathrm{P}(\mathbf{y}|\mathbf{f})$ $\hfill \triangleright$ marginal likelihood bound
14: $\quad$ **return** $\mathbf{f}, \boldsymbol{\alpha}, Z$

---

An important limitation of the Kronecker approach is the fact that $N$ grows exponentially with $d$, which restricts the practical applicability of the algorithm to up to 7 or 8 dimensions. Nonetheless, many nonlinear tuning functions we seek to estimate satisfy this constraint. This problem can be somewhat alleviated by limiting the input set to the biologically-relevant subregion of the hypercube. For instance, in the spatial domain we restrict the test positions to be within the borders of the box (either circular or cross-shaped). A clever discretization of the individual dimensions can also help.

## S3   Comparison of different tuning function estimators

Traditional approaches for estimating nonlinear tuning functions such as place fields rely on histograms, usually followed by smoothing. Despite their appealing simplicity, such estimators require very large amounts of data [5]. Furthermore, the degree of smoothing is usually decided arbitrarily, with potentially misleading results, for instance generating place-field like structure out of noise in the limit of very little data or very low firing rates.

Alternative methods increase estimation efficiency by a probabilistic treatment, that also provides a principled way for setting smoothing parameters. These benefits come at the cost of a significant increase in computational time. Several algorithms of this class have been proposed in the literature, which differ mainly in their treatment of the tradeoff between model flexibility and computational tractability. At one extreme, [5] makes strong assumptions about the covariance structure (sparse band-diagonal inverse covariance, with dependences restricted to nearest-neighbour interactions) but with the fastest implementation (linear in $\mathbf{N}$). At the other extreme, [6] enforces no constraints on $\mathbf{K}$, with cubic computational cost, but relies on an active learning framework to make sure that the very limited data it uses is most informative. Our approach sits between these two in that it achieves

almost linear computational cost ($N = 10^5$ learning done in minutes) but with much more flexibility in terms of the functional structure captured by the prior, especially when combined with expressive kernels such as SM. To provide some intuition for the flexibility of different priors, we show a set of tuning functions drawn from a log-normal Cox process with different covariances (Fig. S1). The Kronecker kernel is as expressive as the standard SE kernel (they are formally equivalent), with different hyperparameters capturing different spatial scales and degrees of smoothness. In contrast, Rad-like covariances have inherently small length scales (due to the nearest-neighbour correlations). Lastly, the Kronecker-SM kernel can represent a combination of structures at different spatial scales.

## Footnotes

[1]To simplify notation, we omit the $\theta$ dependency here.

[2]While we use an uninformative prior for $\theta$ here for simplicity, it may be useful to extent this to a hierarchical version with shared hyperparameters across simultaneously recorded cells or alternatively within cells across sessions.

[3]$\mathbf{H}$ is diagonal, so the only eigen-decomposition that matters computationally is that of $\mathbf{K}$.

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

Figure S1: **Flexibility of different kernels.** Top row: Sample tuning functions $\boldsymbol{\lambda} = \exp(\mathbf{f})$ on a $24 \times 24$ grid, with $\mathbf{f}$ drawn from a GP prior with zero mean different covariances. From left to right: band-diagonal inverse covariance (parameters as in Fig.2 from [5]), traditional SE kernel with $\rho_1 = \rho_2 = 0.3$, and $\sigma_1 = \sigma_2 = 0.5$, Kronecker SE with same parameters; Kronecker spectral mixture kernel with 5 components; weights, spectral means and variances drawn independently from uniform distribution, with weights further normalised to L1 norm 1. Bottom row: corresponding covariance structure (first $200 \times 200$ elements of $K$.)