[Reviews · NeurIPS 2016]

Reviewer 1

Summary

Here the authors present results fitting GP models to hippocampal place fields. These are elegant results that take advantage of recent speed-ups from Kronecker methods. Although this is more of an application paper than a methods paper, it is clearly written and very nicely illustrates how GP methods could become a new standard for tuning curve estimation.

Qualitative Assessment

The importance/effect of the Kronecker assumption could use some additional explanation. It took me some time to figure out that the input dimensions had to have separate kernels, and I'm still not sure exactly what effect this has on the fits. It would be helpful to see some direct model comparisons– e.g. held-out log likelihood for histogram vs GP vs GP+non-spatial - to know how much the non-spatial terms contribute and also to see exactly how the GP out-performs other methods in the low-data limit. It may also be worthwhile to mention/compare parametric models of place fields (e.g. Brown et al. 1998) – several of the cells in Fig 3, for instance, don’t seem to deviate too much from a 2d Gaussian. typo in abstract: “other cortical regions” -> other brain regions

Confidence in this Review

2-Confident (read it all; understood it all reasonably well)


Reviewer 2

Summary

This is a solid paper. The authors extend previous Gaussian process (GP) methods for estimation of multidimensional response functions by exploiting recent advances in scalable inference in GPs with Kronecker product kernels. The proposed method seems to be lifted essentially whole from a recent paper (Flaxman et al ICML 2015), so there is not much of a general machine learning advance here – but the authors seem to have chosen the right tools for this application, and they obtain some nice results by applying the methods to analyze some data from hippocampal neurons. The paper is quite clearly written and the results presented nicely.

Qualitative Assessment

I like this paper, as indicated above. (My scores of 3 should be interpreted as "high 3's.") I just have a few more comments to help the authors clarify their contributions and make the methods more useful for the target neuroscience community. The methods are explained very sketchily - largely by pointing to previous work (largely in conference papers which are themselves fairly concise / sketchy). I'd like to see an appendix added with the full methods explained in detail in a self-contained way, so that computational neuroscientists can understand (and potentially help to extend) the methods. I assume the authors plan to release their code publicly; they should say so explicitly. Bonus points for sharing the link during the author response period, so the reviewers can kick the tires more thoroughly. How are posterior standard deviations of the estimated tuning fields computed efficiently here? Add some details on this. I found fig 2 a bit over-idealized; I'm not sure how many neuroscientists would trust the results in the rightmost panels without further experimental validation. The conclusions of fig 3f were a bit unclear; I might drop these analyses in order to give more space to adequately describe / discuss the other results.

Confidence in this Review

3-Expert (read the paper in detail, know the area, quite certain of my opinion)


Reviewer 3

Summary

The paper describes a GP-based method for inferring the multi-dimensional tuning properties of place cells and grid cells, using expressive spectral mixture kernels to model periodicity, and kronecker methods for computational efficiency.

Qualitative Assessment

The main novel contributions of this paper are the use of spectral mixture kernel functions to model periodicity, Kronecker methods for computational efficiency, and the application of this model to simulated grid cells and real place cells. It should be noted that this is NOT the first paper to apply GP-Poisson modeling to neural data, NOR the first paper to scale beyond 2d input spaces, as claimed in the introduction. In fact, M. Park, J. Weller, G. Horwitz, and J. Pillow, Neural Computation 2014 develop and apply GP-Poisson models to estimate the 3d color contrast tuning of V1 neurons. This paper should be cited here. The use of Kronecker methods requires that the input space be discretized. It seems to me like this might pose problems to scaling this approach to much higher dimensional input spaces, as much of the input space will only be sparsely sampled, yet must be explicitly represented and computed. In other words, doesn't the number of "samples" on the discrete lattice: N, have to scale exponentially with the dimension: d? It would also be nice to see how much faster the Kronecker method is, over a naive non-Kronecker implementation, especially when the input space is very sparsely sampled.

Confidence in this Review

2-Confident (read it all; understood it all reasonably well)


Reviewer 4

Summary

The Authors develop a new method for inferring receptive fields (or tuning curves) given neural counts and multidimensional covariates. This method is essentially an extension of [1], featuring two novelties: i) the use of recent methods in the scalable GP literature (e.g. Kronecker products, [2]) to retain tractability when extending from the 2D case in [1]. ii)an involved choice of the Kernels -- also based on recent observations in the GP literature [3] -- to enhance the ability to extrapolate at subsampled regions. The authors show results based on synthetic and real hippocampus data with behavioral (navigation data) and neurophysiological (network state) covariates. Their results overall reveal that the algorithm does a good job (compared to naïve estimates) and that the properties of the inferred receptive fields is consistent with some recent findings in neurobiology. Further, the authors utilize their method to probe into a detailed neurobiological question; namely, whether or not receptive fields of hippocampus neurons are invariant to direction of motion in a 2D task, and their finding (answer:no) contrast what is understood in 1d navigation task (answer:yes) [1]Rad, K.R. & Paninski, L. Efficient, adaptive estimation of two-dimensional firing rate surfaces 288 via Gaussian process methods. Network 21, 142–168 (2010). [2]Saatçi, Y. Scalable inference for structured Gaussian process models. PhD thesis, Cambridge 290 University, UK, (2012). [3]Wilson, A., & Adams, R. Gaussian Process Kernels for Pattern Discovery and Extrapolation. 297 arXiv.org (2013).

Qualitative Assessment

Overall I think it is an excellent, elegant paper, which also is written with good style and clarity (however, notice the typo in line 111). The authors do a very good job in utilizing recent ML advances to introduce significant methodological innovations in a field. Namely, there is a need for the developing of kernel-based (beyond linearity) smoothing methods for estimation in receptive fields, but these can grow computationally prohibitive as the number of dimensions increases. By appealing to the (recent) scalable GP literature the authors succeed in achieving this goal. I do have a criticism -- based on a very personal opinion -- though: for me it is a bit 'dangerous' to utilize this short-length publication outlet for the communication of new neurobiological findings. The authors do this (lines 225-240 , commenting results on figure 4 ). These findings are based in the method they are just introducing, which immediately raises the question on whether or not they may be spurious, induced by the modeling assumptions. In this vein, I would have deferred the new neurobiological claims for a more specialized publication and instead focus this publishing effort in a more solid elaboration of the method, discussing for example the rationale for choosing only stationary kernels, or for considering kronecker products. Are these reasonable assumption? is there anything lost by choosing kronecker products ? what would happen if the receptive fields were intrinsically non-stationary? does this happen in reality? Also, I would have liked to see a comparison to more sophisticated estimators beyond the saturated, empirical estimator (e.g. how does your method compare to [1]?) But again, besides this personal opinion it is a great paper (and particularly enjoyed figure 2). [1] Rad, K.R. & Paninski, L. Efficient, adaptive estimation of two-dimensional firing rate surfaces 288 via Gaussian process methods. Network 21, 142–168 (2010).

Confidence in this Review

2-Confident (read it all; understood it all reasonably well)


Reviewer 5

Summary

In their paper "Estimating nonlinear neural response functions using GP priors and Kronecker methods", the authors apply Gaussian process priors to estimate multivariate tuning functions. They test their methods on artificial data with known ground-truth and apply their methods to real experimental recordings, confirming and extending findings in the hippocampal system.

Qualitative Assessment

For the most part, the method validation is appropriate. As the authors point out, the ground truth statistics exactly follow the method assumptions. Nevertheless, the authors also show on real data that their results are consistent with traditional place fields which aids in giving their method credibility. I don't think the comparison with traditional histogram based estimates is very telling though (line 138, Fig 1b). One could do much better by just applying simple smoothing. The GP formalism with learning using Kronecker methods is not new, but its application to estimate multivariate response functions is novel, as far as I can tell. The use of spectral mixture kernels in particular has great potential for interesting applications. System characterization by means of tuning functions is a very widely used technique in neuroscience. The limited number of samples that are typically available in neuroscience applications makes it challenging to estimate these functions, in particular if many variables are involved. Leveraging modern GP techniques to overcome these difficulties is a significant contribution that may find applications beyond the hippocampal system which is analyzed in this paper. I think the underlying assumptions of GPs in conjunction with the inhomogeneous Poisson process should be layed out more clearly in the paper (the authors very briefly mention smooth dependence of the firing rate on the input in line 71). Practitioners who just want to apply the method should be able to get an idea of its limitations. Otherwise, the paper is very clearly written. All figures are intuitive and complement the text very well. The level of mathematical details is appropriate for the presented methods. Typos: Line 101: it -> its Line 111: with -> which Line 145: they reasonably -> they were reasonably Line 186: increases -> increase Line 234: directions -> direction Update: After reading all reviews and the rebuttal, I still think my scores are appropriate.

Confidence in this Review

2-Confident (read it all; understood it all reasonably well)


Reviewer 6

Summary

The authors make use of a recently developed and fast variational algorithm to learn Gaussian processes (GP) from data with non-Gaussian likelihoods to learn the tuning properties of neurons in hippocampal cells. After quickly introducing the mathematical details on which the main algorithm is based, the authors show the robustness of their estimators on simulated data. Finally, they apply the technique on data from hippocampal area AC1 and discuss their findings.

Qualitative Assessment

In this work the authors make use of an already developed technique, so for this reason I don't find the paper innovative on the technical side. The authors emphasize that the focus is on the application of the method to infer multi-dimensional tuning functions of hippocampal neurons (line 40). Nevertheless they don't try to infer tuning functions in more than 3 dimensions (2 spatial + 1 non-spatial variables), and in my opinion this reduces the potential novelty that their results may have. In the next paragraphs I give a detailed opinion about the content and the structure of the paper. Content Subsection "Detailed parameter settings for experiments", page 4. The authors may consider to mainly reorganize this section, maybe also to remove it completely from here and merge it with the section in which they talk about the application of the method of neural data (see the following points in my review). Section 3 "Results": In my opinion, there should be a clear separation between the estimator validation section (which may include all the experiments run on simulated data) and the section where the method is applied to data from neural recordings. Subsection "Spectral mixture kernels for complex functional dependencies", page 5. I am not sure about the usefulness of the first experiment (extrapolating the tuning function outside rectangular region): the fact that using a spectral mixture kernel make it possible make extrapolations was already a well established fact in reference [9], and I would be surprise if this was not the case in this simple experiment. Rather, I would suggest to focus more on the second experiment, in which learning a correct representation of the tuning function is not trivial at all. Subsection "Spatial and non-spatial modulation of CA1 responses": The authors may consider to reorganize the structure of this subsection, which represents the very core of the paper. A brief description of the data should be given at the very beginning of the paragraph (experimental techniques, number of recorder cells, few words about preprocessing the raw signals). I noticed that some of this information is already reported, but spread across the whole section, making it difficult for the reader to have a clear picture of what kind of data was used. An important analysis that in my opinion is missing in this sections is the estimation of the full joint posterior distribution over all the spatial and non-spatial variables at the same time. Only multiple 3D fields consisting of the spatial location plus an additional non-spatial variable are estimated. The authors claim that this choice was made for the sake of simplicity (Footnote #11), but the focus of the paper is "not on the methods per se but rather on their previously unacknowledged utility for estimating multidimensional nonlinear tuning functions", as stated by the authors (line 40). Therefore it is really important to see these methods applied to data which is really multidimensional (with dimension larger than 3). In this case, the 3D fields estimation would still be useful, using their predicted marginal posterior distributions to assess the quality of the the marginal posterior distributions predicted by the full multidimensional field. General questions/remarks I did not understand why the corners are missing in the plots in Figure 1d. It looks like this plots correspond to the walls of a box that was open and flatten on the plane, somehow corresponding to a 2D representation of a 3D plot. Is this the case? Please consider to specify it in the caption of the figure or to change the shape of the plot, because, as it is now, it is difficult to understand. The authors should consider to reduce the number of footnotes. Overall, there are 14 footnotes, which sometimes contain also important information (#11, #12, #14). Page 4, line 126: the authors refer to a parameter "k", stating that it takes discrete values. Nevertheless, the meaning of "k" is not defined until line 176. Page 4, line 135: a reference would be important here. On page 7: several correlations between pairs of variables are reported. In all cases, brackets are misaligned, and the last one is always missing (lines 207, 208, 211, 221, 222).

Confidence in this Review

3-Expert (read the paper in detail, know the area, quite certain of my opinion)